# Ergodic seismic precursors and transfer learning for short term eruption forecasting at data scarce volcanoes

Alberto Ardid [1,18] ✉, David Dempsey [1,18], Corentin Caudron [2,3], Shane Cronin [4], Ben Kennedy [1], Társilo Girona [5], Diana Roman [6], Craig Miller[7], Sally Potter[7], Oliver D. Lamb [7], Anto Martanto[8], Yesim Cubuk-Sabuncu[9], Leoncio Cabrera [10], Sergio Ruiz[11], Rodrigo Contreras [12,13], Javier Pacheco[14], Mauricio M. Mora [15] & Silvio De Angelis [16,17]

Seismic data recorded before volcanic eruptions provides important clues for forecasting. However, limited monitoring histories and infrequent eruptions restrict the data available for training forecasting models. We propose a transfer machine learning approach that identifies eruption precursors—signals that consistently change before eruptions—across multiple volcanoes. Using seismic data from 41 eruptions at 24 volcanoes over 73 years, our approach forecasts eruptions at unobserved (out-of-sample) volcanoes. Tested without data from the target volcano, the model demonstrated accuracy comparable to direct training on the target and exceeded benchmarks based on seismic amplitude. These results indicate that eruption precursors exhibit ergodicity, sharing common patterns that allow observations from one group of volcanoes to approximate the behavior of others. This approach addresses data limitations at individual sites and provides a useful tool to support monitoring efforts at volcano observatories, improving the ability to forecast eruptions and mitigate volcanic risks.

Volcanic eruptions are one of the most spectacular natural phenomena on Earth, but they can be deadly if not properly anticipated. Eruption forecasting is the practice of estimating the likelihood, location, and timing of eruptions[1]; and the type of hazards they can present[2]. Volcano observatories are a key line of defense to ensure the safety and welfare of the ~29 million individuals who reside within 10 km of active

volcanoes[3]. Eruptions are also disruptive to air travel and transportation systems, regional air quality, food and drinking water systems, and even influence weather patterns[4]. Several recent eruptions have been successfully forecasted due to effective monitoring and alert systems[5], including 1991 Mount Pinatubo[6] (Philippines, by PHIVOLCS and USGS), 2009 Redoubt[7] (Alaska, USA, by AVO), 2010 Merapi[8] (Indonesia, by

[1]University of Canterbury, Christchurch, New Zealand. [2]Université libre de Bruxelles, Brussels, Belgium. [3]WEL Research Institute, Brussels, Belgium. [4]University of Auckland, Auckland, New Zealand. [5]Alaska Volcano Observatory, Geophysical Institute, University of Alaska Fairbanks, Fairbanks, AK, USA. [6]Carnegie Institution, Washington, DC, USA. [7]Te Pū Ao | GNS Science, Taupo, New Zealand. [8]Center for Volcanology and Geological Hazard Mitigation, Bandung, Indonesia. [9]Icelandic Met Office, Reykjavík, Iceland. [10]Departamento de Ingeniería Estructural y Geotécnica, Pontificia Universidad Católica de Chile, Santiago, Chile. [11]Departamento de Geofísica, Universidad de Chile, Santiago, Chile. [12]Departamento de Geología, Universidad Católica de Temuco, Temuco, Chile. [13]Centro de Investigación en Evaluación de Riesgos y Mitigación de Peligros Geológicos, Geokimün, Facultad de Ingeniería, Universidad Católica de Temuco, Temuco, Chile. [14]National University of Costa Rica, Heredia, Costa Rica. [15]Central American School of Geology, University of Costa Rica, San Jose, Costa Rica. [16]University of Liverpool, Liverpool, UK. [17]Istituto Nazionale di Geofisica e Vulcanologia, Pisa, Italy. [18]These authors contributed equally: Alberto Ardid, David Dempsey. ✉e-mail: aardids@gmail.com

CVGHM), 2014 Villarrica[9] (Chile, by OVDAS), 2021 La Soufrière (in Saint Vincent and the Grenadines, by UWI-SRC) and 2014 Bárðarbunga[10] (Iceland, by NCIP) eruptions. These cases exemplify the primacy of scientific monitoring and underscore the importance of historic data for effective eruption forecasting.

Eruption forecasting frequently relies on the detection of meaningful signals encoded in seismic data, i.e., eruption precursors. Seismic amplitude filtered to informative frequency bands is often used, e.g., real-time seismic amplitude measurement[2,11] (RSAM) and displacement seismic amplitude ratio[12,13] (DSAR). Precursor signals[14] are thought to reflect fluid-rock interactions in magma plumbing and hydrothermal systems[12,13], induced oscillations in cracks and conduits[15,16], or accumulations of pressurized gas[12,13]—all physical processes that may precede an eruption.

For many volcanoes, eruptions are infrequent events that punctuate an otherwise calm state of repose. Although modern seismic monitoring and data archiving were pioneered over the last half-century, it has only proliferated in recent decades. Excluding persistently active examples, e.g., Etna (Italy), Stromboli (Italy), Fuego (Guatemala), most volcanoes have few or only one eruption on record. For example, of the 24 volcanoes considered in this study, only 3 have more than 3 eruptions, and 11 have only one seismically recorded event. Here, we define an eruption as a discrete period of volcanic activity characterized by the deposition of new volcanic material outside the vent, which may include lava, ash, or pyroclastic material. Emissions of gas alone are considered precursory activity unless accompanied by the ejection of other materials.

The relative scarcity of eruption data makes data-driven forecasting challenging, leading to the use of generalized forecasting models that apply insights across multiple volcanoes. Such models leverage the similarities between volcanoes, reducing the need for extensive customization to each target volcano[17–20]. However, they face challenges due to the varied types of volcanic systems and their range of precursory behaviors. Unheralded eruptions with subtle or no known precursors[2,21] further complicate forecasting efforts. Typical approaches within generalized forecasting include analog methods[15,16], heuristic elicitation methods[22], physics-constraint methods[23], and inter-event models[24,25]. Analog approaches assess hazards by comparing less-documented volcanoes to better-studied analogs, though they rely on expert judgment and may lack consistency across cases. Heuristic elicitation methods similarly draw on expert insight for rapid assessment but remain inherently subjective. Inter-event models, which use historical eruption intervals to estimate future eruption likelihood, provide a quantitative basis but are often less effective for volcanoes with irregular activity patterns. Also, catalog gaps—due to data scarcity, limited investigation, or natural erosion of smaller deposits—introduce uncertainty in recurrence estimates, especially for smaller events.

In this context, machine learning (ML) has emerged as a tool for extracting forecasting information from large and complex volcanic datasets[26–28]. While it has the potential to mitigate human limitations in assessing precursors[22], it risks introducing its own biases via eruption labeling, data curation, and algorithm selection[29,30]. Nevertheless, its ability to integrate varied data types, including seismic[31,32], gas[33], and geodetic data[34], opens avenues for improved characterization of volcanic activity including forecasting.

In this work, we used transfer machine learning to identify sets of precursor signals that are shared amongst groups of volcanoes and to evaluate the predictive skill of these precursors in simple forecast models. We used time series feature engineering[22,28] to extract statistically significant patterns in seismic data across multiple volcanoes. Our dataset comprises 41 eruptions across 24 volcanoes (Fig. 1a) with a combined seismic record length of ~73 years. Precursors are identified a fix time window on their higher rate of recurrence prior to eruptions and reduced frequency otherwise while implementing safeguards to

minimize false discovery. Precursors extracted from a volcano pool were then used to train forecasting models, which were later tested on unobserved target volcanoes outside that pool. We developed three different generalized forecast models based on eruption type: (1) a magmatic model, involving 9 volcanoes and 16 eruptions, (2) a phreatic model, involving 6 volcanoes and 15 eruptions, and (3) a global model, involving all volcanoes and eruptions (Fig. 1b). Random forest models were trained with 48 h backward looking windows to be especially sensitive to rapid volcano changes that requires responsive monitoring. For a given instance in time, the output of a model is a non-probability value between 0 and 1, with higher values indicating an increased likelihood of eruption based on the previous 48 h of seismic data. All models were assessed in terms of their out-of-sample performance, measured on data from test volcanoes that were withheld during feature selection and training. This pseudo-prospective forecast, which simulates real-time conditions by withholding target data during training and testing it as unseen data, is not as rigorous as true prospective forecasting that evaluates future, genuinely unseen data. However, it is superior to hindcasting models in its ability to avoid overfitting by not allowing the model to train on the same data it is tested on, providing a more realistic assessment of the model's performance on unseen volcanoes. Forecasts were constructed for the entire 73-year dataset, with each volcano taking its turn in the test set.

## Results
### Forecast performance and benchmarks
In this research, a good forecasting model is one that outputs a strong positive response (values approaching 1) in the 48-hours before an eruption and a relatively depressed response (values closer to 0) during the extended periods between eruptions. The discriminability of these models is quantified by their Receiver Operator Characteristics (ROC) curve and associated Area Under the Curve (AUC). AUC values approaching 1 have a negligible false positive rate during unrestful episodes[35,36] and no missed eruptions.

ROC curves and AUC values for models constructed from the three volcano pools are shown in Fig. 2a–c against a reference AUC of 0.5, which denotes no discrimination of eruptions. All three models have AUC values of about 0.8, which indicates a modest ability to discriminate eruptions from non-eruptive unrest at the unobserved target volcanoes. However, as the acceptable range of model AUC varies with context and application, it would be premature to claim such a model is good enough for operational use. Nevertheless, we can benchmark this performance against other kinds of forecast model.

Direct measures of seismic amplitude (e.g., RSAM) are commonly used for volcano monitoring due to their high sampling rate and informativity on volcanic processes when combined with complementary metrics, observational data, and interpretative models that provide context and insight into the underlying volcanic activity[11]. Thus, RSAM is a good reference to benchmark the performance of the ML models developed here, as it offers continuous, high-resolution data. In contrast, other monitoring data, such as gas sampling or geodetic observations, often have lower temporal resolution and are scarcer, making them less suitable for direct, consistent comparison. We find that simple RSAM triggering models generally underperform compared to ML models (Fig. 2) even when the RSAM benchmark is afforded the benefit of hindsight by fine-tuning its parameters for maximum sensitivity (an advantage not granted to the ML models). Seismic amplitude models are most sensitive to phreatic eruptions (AUC = 0.74, Fig. 2b), though less sensitive than the corresponding ML model (AUC = 0.8). Average AUC values are depicted in Table 1. This indicates that RSAM-based forecasting in our dataset shows some dependence on eruptive style. However, it is important to clarify that these findings are based on a limited number of case studies within our dataset. The seismic model had negligible forecasting value on the

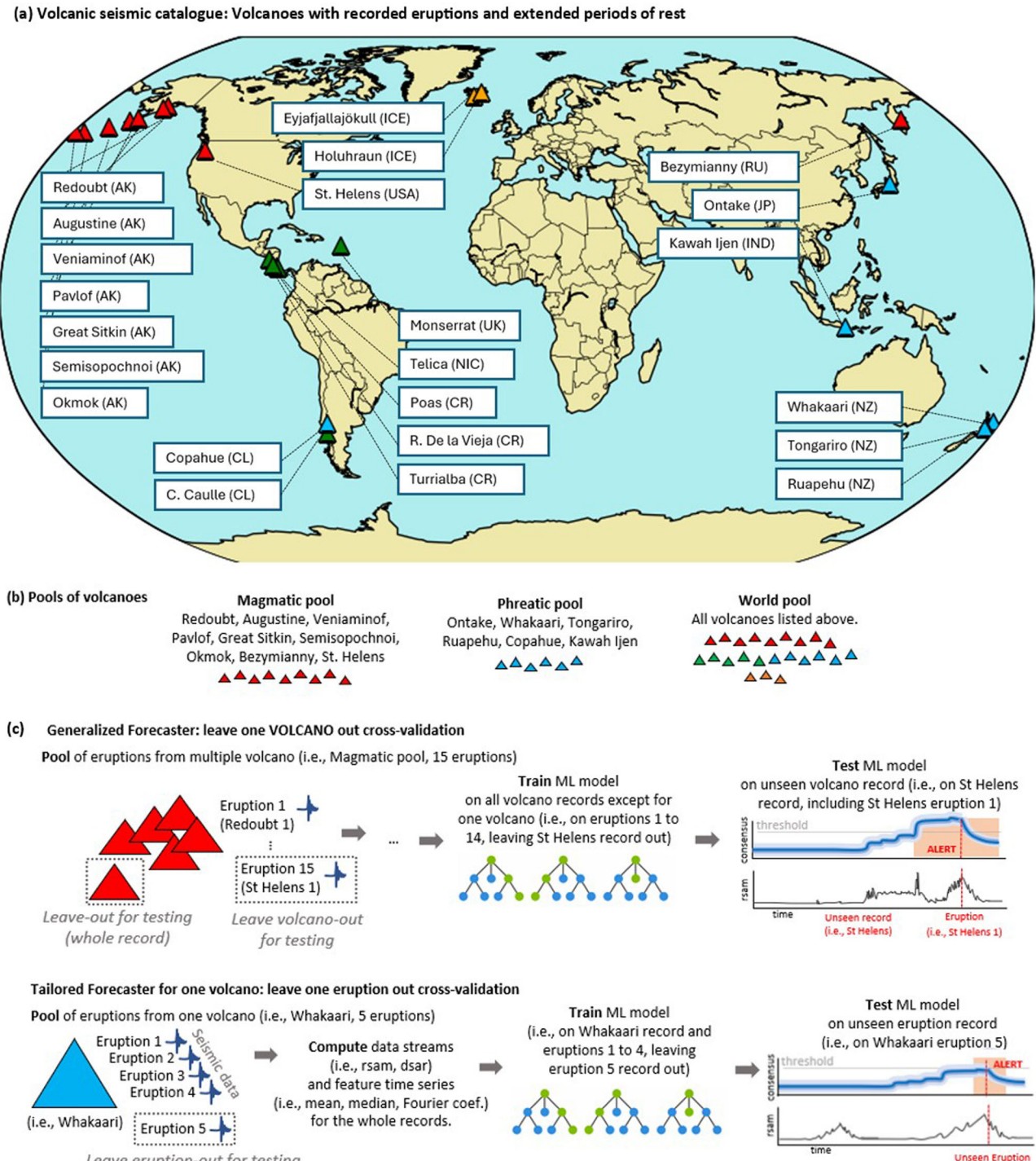

**Fig. 1 | Catalog of volcanoes, eruption groupings, and cross-validation strategies for forecasting models. a** Map of volcanoes and eruptions used in this study. **b** Sets of volcanoes grouped to train generalized eruption forecasting models (pools). **c** Cross-validation testing strategies. Generalized forecasters leave out an entire volcano's record for later testing, whereas tailored forecasters leave out only one eruption at a time. Further cross-validation details are given in Methods.

magmatic pool (AUC = 0.5, Fig. 2a). This comparison is not intended to denigrate the value of RSAM in volcano monitoring, which is consistently demonstrated at observatories around the world. Instead, we argue that the latent patterns embedded within RSAM, and other seismic intensities are more reliably extracted with a transfer machine learning approach.

We also compared the performance of generalized ML models to tailored alternatives, i.e., ML models trained exclusively on individual

volcanoes (Fig. 2d–f). This lets us assess the value of transferring precursors between volcanoes, independent of the ML pipeline itself. Bezymianny, Whakaari and Copahue volcanoes are three examples with enough eruptions for a meaningful comparison. Both tailored and generalized forecast models have comparable discriminability, as indicated by similar AUC values. This provides further evidence that seismic precursors can be transferred to unobserved target volcanoes with minimal loss of forecasting skill.

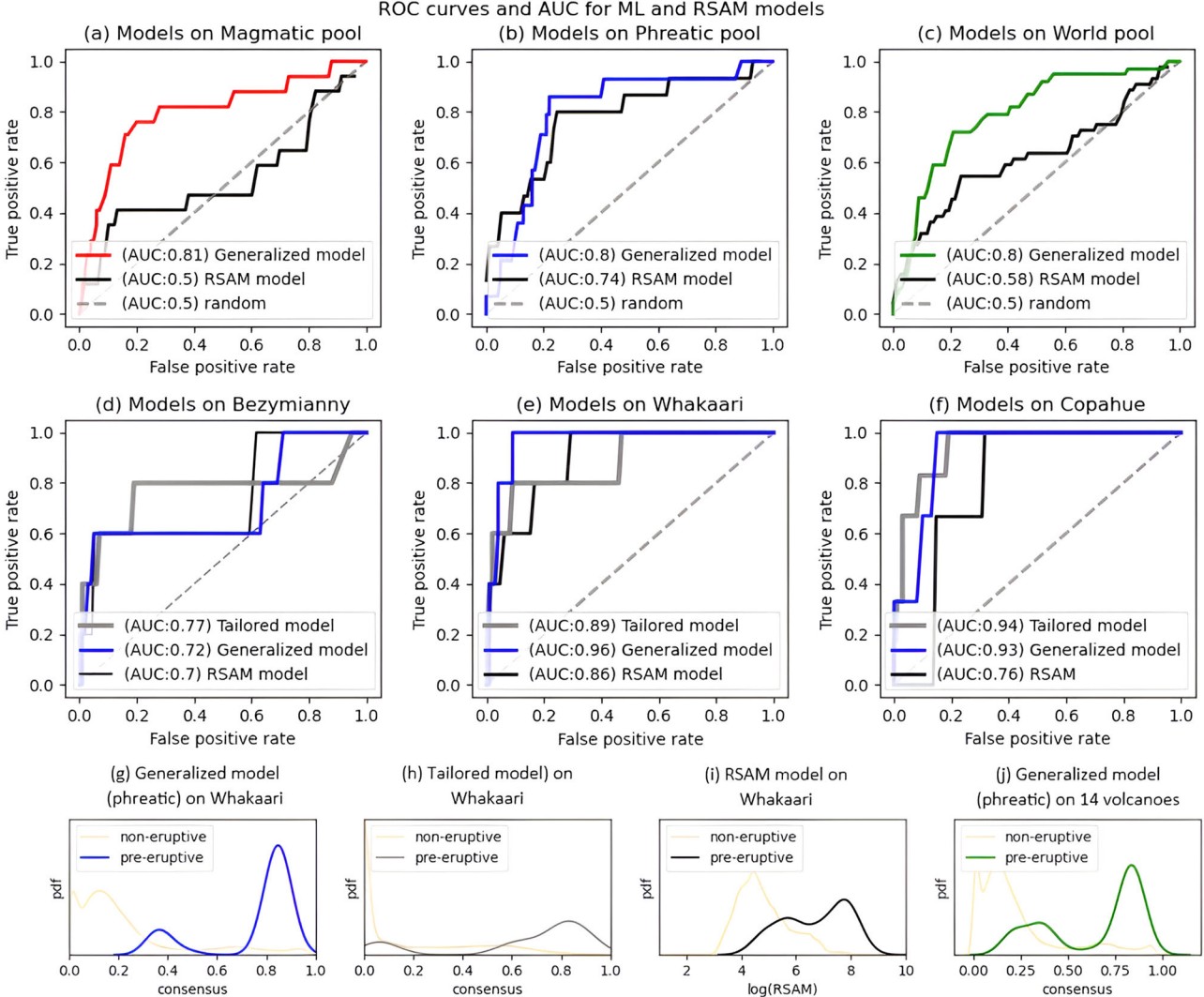

**Fig. 2 | Relative performance of generalized and tailored machine learning forecasts on out-of-sample volcano data and benchmarked against seismic intensity (RSAM) models.** ROC = Receiver Operating Characteristic curve and AUC = Area Under the (ROC) Curve are metrics that quantify a model's ability to discriminate pre-eruptive signals from the non-eruptive background. Subplots (**a**–**c**) show performance of the three generalized forecasters (magmatic, phreatic, and world pools). Subplots (**d**–**f**) compare performance of tailored and generalized forecasters for Bezymianny, Whakaari, and Copahue. Diagonal dashed lines show a reference random model with no predictive skill. Models with higher AUC have greater predictive skills. (**g**–**i**) Relative frequency of forecast values, distinguishing between pre-eruptive and non-eruptive windows, over the 10-year record at Whakaari (Fig. S2). A larger separation between distributions denotes improved predictive skill. **j** The same frequency plot as (**g**–**i**) but for 14 volcanoes in the phreatic pool.

## Ergodic characteristics of eruption precursors

The predictive skill of generalized forecasting models depends on the number of volcanoes in their training pool. Models trained on one or two volcanoes tend to have highly variable but relatively low performance (Fig. 3) when making out-of-sample predictions (0.35 < AUC < 0.65). Transfer learning is strengthened for at ensemble sizes larger than three but appears to saturate for models with more than twelve volcanoes. Performance does not increase substantially once an AUC of around 0.8 is reached, suggesting an upper limit on the predictive skill. Thus, careful consideration should be given to the relative costs and benefits associated with dataset expansion, particularly when the marginal gains are diminishing over time. Improvements to the way pre-eruptive information is discriminated and used within the model architecture may be needed to realize further scalability.

The generalized models explored here exploit ergodicity in volcanic systems, specifically at a time series feature level (Fig. 4). Ergodicity implies that the distribution of volcano signals over time at a single site can approximate the distribution of signals across a sufficiently large ensemble of volcanoes. This characteristic allows observations from multiple volcanoes to approximate the long-term behavior of individual volcanoes. Without this ergodic property, the predictive improvement observed with larger training ensembles would not be expected. Such characteristics have been demonstrated for satellite-based monitoring of deformation prior to eruptions at 540 volcanoes[37]. However, geodetic data on its own has limitations, with no eruption occurring at nearly half the deforming volcanoes[37]. Here, we estimate that ensembles with as few as twelve volcanoes are likely capturing a large proportion of the common seismic eruption precursors.

## Practical challenges for operational forecasting

Demonstrating the forecasting skill of transferred eruption precursors is a necessary but not sufficient step for their operational use. Volcano monitoring scientists routinely integrate a wide range of seismic[31], thermal[32], gas[33], and geodetic data[34] across disparate time scales[13,28,38,39] and use these to inform experts mental models of the evolving

**Table 1 | AUC values for the forecasting models depicted in Fig. 3 (ML: Machine Learning; RSAM: Real Time Seismic Amplitude Measurement for 6 h average)**

| AUC | Whakaari | Bezymianny | Copahue | Magmatic | Phreatic | World | Mean |
|---|---|---|---|---|---|---|---|
| ML tailored | 0.89 | 0.77 | 0.96 | – | – | – | 0.87 |
| ML Generalized | 0.96 | 0.75 | 0.93 | 0.81 | 0.80 | 0.80 | 0.84 |
| RSAM 6 h | 0.86 | 0.70 | 0.76 | 0.50 | 0.74 | 0.58 | 0.69 |

Forecaster models include tailored models for Whakaari, Bezymianny, Copahue, as well as Generalized models for the Magmatic, Phreatic, and World pools. The "Mean" column is the average AUC across all models.

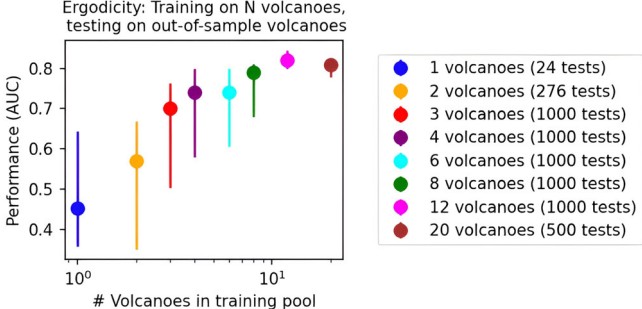

**Fig. 3 | Performance gain of generalized forecast models as the size of the volcano ensemble is increased.** Median and 33-67 percentile range of AUC for models trained with ensemble sizes ranging from 1 to 20 volcanoes. Ensemble membership is selected randomly from the world pool, whereupon a train a 25-decision tree model is trained and then tested on the remaining unselected (out-of-sample) volcanoes in the pool. Performance appears to saturate at an AUC > 0.8 for ensembles sizes larger than 8 volcanoes.

volcanic hazard[6–8,10]. If they are to provide complementary input to this process, forecasting models need to be interpretable on a real-time time-series basis. Although this problem remains outstanding, a presentation of individual forecast characteristics is useful here to define its challenging aspects.

The ultimate objective of generalized forecasters is to identify eruption precursors with the same or higher confidence than corresponding tailored models. This is exemplified in the case of the 2009 Bezymianny and 2016 Whakaari eruptions (Fig. 5a, b) with strengthening generalized forecast output prior to those eruptions. The latter event has been particularly challenging to anticipate in prior studies[28] that relied on tailored models, apparently confirming an advantage of transferred eruptions precursors. In other cases, we confirm that introduction of a generalized model has not compromised performance of the tailored alternative, e.g., the 2019 Whakaari eruption (Fig. 5c).

However, such improvements are not realized across all volcanoes. For example, at Copahue (Fig. 5d), generalized models produce consistently elevated outputs between closely spaced events (~weeks), whereas the tailored models do not. Thus, a challenge remains that, while generalized approaches may increase average forecasting skill, they may in some instances produce worse accuracy than tailored alternatives. At Copahue, we hypothesize that this arises because training eruptions at other volcanoes were relatively large compared to the locally reported minor eruptions tested at Copahue[40]. Thus, this volcano may benefit more from a model specially tailored.

The 2010 eruption of Eyjafjallajökull (Fig. 5e) illustrates a challenge of eruption variety. Generalized forecasts were ambiguous before the initial fissure events (reported as the main event in Table S2), which could be due to the closed-conduit nature of the eruption[14]. However, the forecasts did show improved sensitivity to subsequent explosive events, which were open or semi-open conduit eruptions[14]. While this accords with this study's focus on training to

recognize explosive events at other volcanoes, it does highlight the narrow skill of the resulting model. Future models might fine-tune on, or exclusively train with, fissure-type events, or make use of an open/closed conduit subclassification. However, this would necessitate more training data and may require upstream model changes like longer pre-eruptive windows to capture magma ascent processes.

We also report some genuine failures, where generalized models are entirely insensitive to pre-eruptive processes, e.g., the 2011 Cordon Caulle eruption (Fig. 5h), which was weakly signaled by geodetic observations[41], marked by a very small uplift prior to the eruption. This highlights how generalized models can only complement, not replace, other monitoring activities and that weighing conflicting evidence is likely to be an ongoing challenge for monitoring scientists. In this particular case, the missed eruption could be due to the shortened monitoring record that begins only 20 days prior and was hence unable to establish a baseline, although we cannot exclude that its precursors are genuinely distinct from all others in the training set. Additionally, the station's 10 km crater-station distance may have impacted precursor detection.

Finally, Fig. 5f and g compare forecasts of our generalized models over nine-month periods centered on the 2014 Ontake and 2004 Mount St. Helens eruptions. The 2014 Ontake eruption was largely unexpected[42], as its phreatic nature provided limited precursory signals. The 2004 St. Helens eruption was partially anticipated[43] due to increased seismicity and gas emissions, which enabled early warning and monitoring efforts. Our models predict both eruptions with modest confidence; however, the Ontake forecast consistently issues high values over extensive non-eruptive periods (Fig. 5g), indicating a high rate of false positives. In contrast, the St. Helens model only shows sporadically high values (Fig. 5f), suggesting greater sensitivity to specific unrest and pre-eruptive conditions. Addressing and reducing the high false-positive rate, particularly in cases like Ontake, remains a key area for further refinement.

## Discussion

The primary outcome of this study has been to establish the existence of ergodicity in seismic precursors to volcanic eruptions. Ergodicity implies that the ensemble distribution of volcanic signals across different volcanoes can serve as a proxy for temporal behavior at a single volcano. This finding underpins transferability of eruption precursors, offering a statistically consistent foundation for generalized forecasting approaches. While the ultimate impact of this result may one day be the improvement of forecasting skills, particularly at volcanoes with insufficient historical data, there remain several challenges to be addressed. Additionally, while the model estimates eruption likelihood, it does not provide information on the potential magnitude of the eruption—an area for further research.

Scarcity of eruption data at individual volcanoes is a consequence of the infrequency of eruptions compared to the relatively brief length of the instrumental era. This shortcoming is partially addressed within the volcanic crisis response community through use of analog volcanoes[44,45] that provide a guide for hazard evolution. Generalized forecasting models explored here adopt a similar approach, albeit with

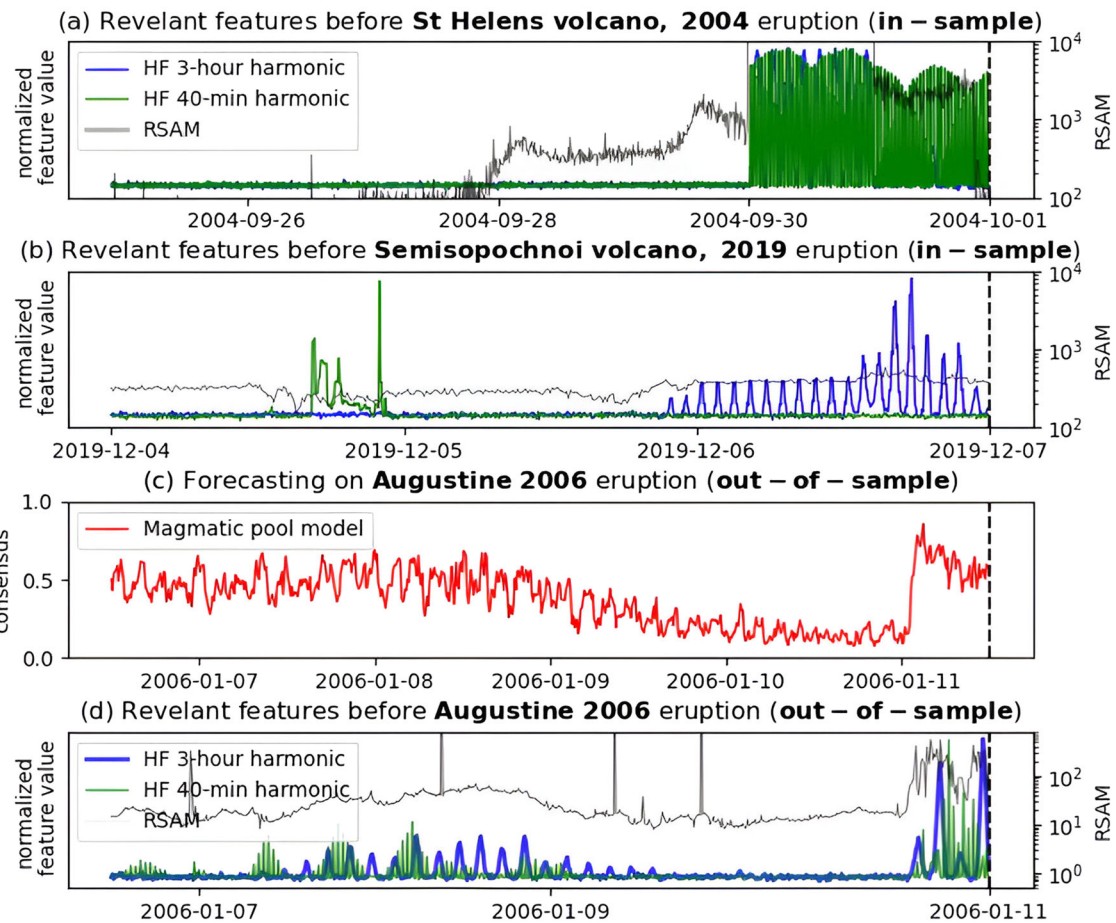

**Fig. 4 | Illustration of precursor identification and transfer to out-of-sample target volcano. a, b** Example of two features identified as significant during model training due to their elevated strength prior to the (**a**) 2004 eruption of St Helen's and (**b**) 2019 eruption at Semisopochnoi: 40 min (green) and 3 h (blue) high-frequency (HF) oscillations. **c** A model trained on these and other features fore-casting on the out-of-sample 2006 eruption at Augustine. **d** The Augustine forecast model strengthens at the same time that the features identified in (**a**) and (**b**) reappear. Note: The ML models developed here exploit ergodicity at the time series feature level. For example, High frequency (HF) seismic oscillations at 40 min and 3 h are discriminated in pre-eruptive sequences from volcanoes in the training ensemble (e.g., Mt St Helens and Semisopochnoi). Later, these same fea-tures are seen to strengthen prior to an eruption at the unobserved target volcano (Augustine). This demonstrates the functional basis of precursor transfer among volcano pools. While ergodicity implies shared physical mechanisms for most selected features, we do not advance a specific hypothesis for the underlying physics of this particular feature. Instead, ergodicity provides a statistical foun-dation for transferability, allowing ensemble-derived precursors to approximate temporal behaviors at individual volcanoes. Nevertheless, we cannot exclude the possibility of spurious correlations in this or other specific cases.

precursor selection and evaluation devolved to an algorithmic level. Essentially, the approach we present here is a form of the analog method; however, rather than relying on expert assessment to select analogs, we use an objective evaluation of seismic patterns across multiple volcanoes to address the issue of data scarcity. By directing analog assessments toward a data-driven approach, we sacrifice some of the expert-driven selection[22,23] in favor of a more statistically con-sistent process, allowing the model to generalize precursor patterns across varied systems. While this affords certain advantages in speed, recall, and objectivity, they are ultimately limited by their narrow focus on seismic data and lack of human reasoning. Nevertheless, such ergodic forecasting models may one day complement monitoring activities at volcano observatories with limited resources or eruption records, with their effectiveness further improved by a deeper con-nection to the underlying physical processes.

We recognize that ergodicity is not a bottomless well. Simply increasing the size of the training ensemble—including more volcanoes and broader behavior—can saturate performance (Fig. 3). In other cases, a larger pool is detrimental, for instance, where better predictive performance is achieved using phreatic and magmatic pools (i.e., Fig. 5). Adding more data to the training pool can introduce noise that

diminishes model reliability. Variability in volcanic data, including sensor limitations, environmental conditions, and data collection errors, add further complexity to signal interpretation. Incorrectly dated eruptions are likely to poison model training. Thus, data cura-tion and ensemble selection are important—but they are also a source of bias. Trade-offs between data diversity and model accuracy will require careful consideration in any operational forecasting application.

Similar ergodic paradigms are exploited in seismic hazard assessment through analysis of large regional earthquake catalogs for frequency-magnitude and ground motion patterns[46,47], and also in flood forecasting through regionalization approaches[48]. Those experiences have foreshadowed similar challenges identified here, including selection, curation, and completeness of the volcano ensemble, and the identification of outlier volcanoes with large errors relative to the ensemble average. These issues may in future be addressed through new model architectures[48], including non-seismic monitoring data (e.g., gas data[33], thermal imagery[32]), physics-informed approaches[23,49], or leveraging machine learning techniques integrated with societal risk considerations to enhance decision-making during volcanic crises[49].

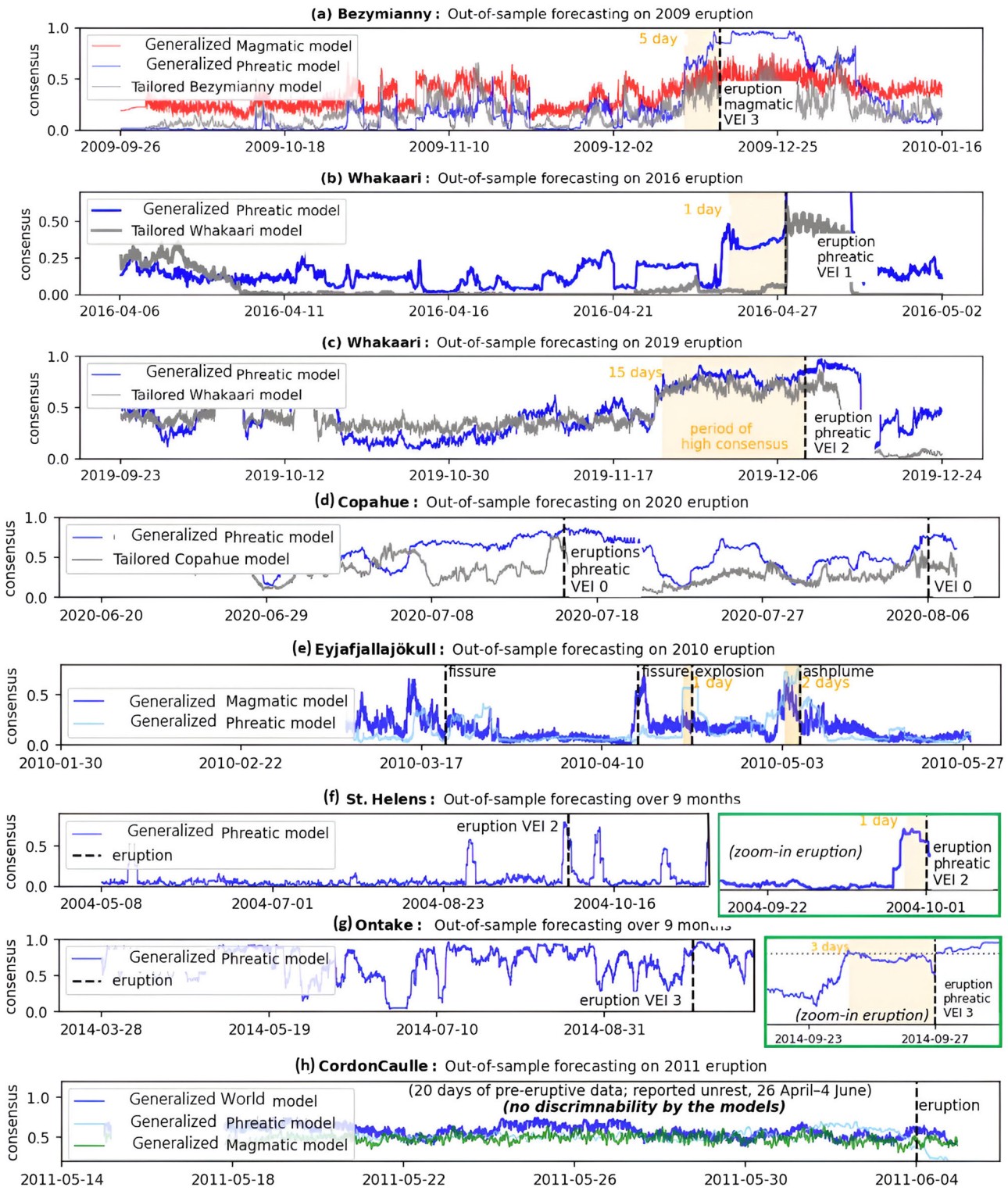

**Fig. 5 | Out-of-sample forecasts from different models prior to eruptions at seven volcanoes.** The figure shows forecasts for (**a**) Bezymianny, (**b**) Whakaari (2016 eruption), (**c**) Whakaari (2019 eruption), (**d**) Copahue, (**e**) Eyjafjallajökull, (**f**) St. Helens, (**g**) Ontake, and (**h**) Cordon Caulle. Pools are indicated in the legend, and eruption times are marked by black dashed lines. The model consensus (solid lines)

is presented as a 2-day rolling 90th percentile of the model output. These cases highlight various practical challenges in applying forecast models, such as differences in eruption style, pre-eruptive patterns, and false-positive rates. Similar plots for additional eruptions are provided in the Supplementary Material (Figs. S4–S7).

## Method
### Machine learning pipeline (workflow)
We used continuous seismic data from 24 volcanoes, with one station per volcano (as shown in Fig. 1a and listed in Supplement Table S2). Where multiple stations were available, we selected the station that

had recorded the most eruptions and was closest to the eruptive vent. Multiple station workflows are not preferred for this study as they narrow the applicability of generalized models to only those volcanoes having similar network characteristics. The dataset used here includes 41 major eruptive episodes. The majority have been classified by the

Global Volcanism Program, 2023. We excluded minor eruptive activity, such as geysering and passive ash emissions, as our focus was on the initial abrupt, impulsive (phreatic and magmatic) onset of eruption. We also excluded effusive eruptive episodes, whose different characteristics might be addressed in future work. The machine learning pipeline[28,50] is summarized below.

## Data preprocessing
We use the vertical velocity component time series at each station with the instrument response removed. We processed the time series from each station to generate four separate data streams, each consisting of its own time series sampled at 10-minute intervals[22,28]. Three data streams are generated by bandpass filtering to three frequency ranges, which captures different parts of the volcano-seismic signal. Filtering between 2 to 5 Hz focuses on a tremor signal of frequent volcanic origin while excluding ocean noise at lower frequencies (mainly <1 Hz). This filtered time series was incremented into 10-min, non-overlapping windows with the average absolute velocity computed for the window. We refer to this as Real-time Seismic Amplitude Measurement (RSAM) although we acknowledge that RSAM is sometimes computed on the unfiltered or very loosely filtered trace, e.g., 0.5–20 Hz to exclude microseism/anthropogenic contributions. We compute Median Frequency (MF) and High Frequency (HF) data streams in the same way as RSAM but instead filtering in the range of 4.5 to 8 Hz and 8 to 16 Hz, respectively. These two data streams focus on signal attenuation effects above the frequency range from which tremor energy is commonly radiated[16]. Finally, Displacement Seismic Amplitude Ratio (DSAR) is calculated as the ratio of the integrals of the MF and HF signals[22]. High values of DSAR have been inferred to correlate with high gas levels in the edifice, suggesting either reduced fluid motion and/or trapping that has led to a gas-accumulation[38,51]. Time series gaps were imputed by linear interpolation and because these gaps occur in noneruptive periods, a later step of downsampling reduces their impact on the models[50].

## Regional earthquake filtering
To isolate continuous volcanic signals, we removed signatures from regional and volcano-tectonic (VT) earthquakes. Wave trains from earthquakes perturb time-averaged station velocity above the background, which then confuses ML algorithms that don't have the context of human operators. To identify and remove earthquake effects, we applied outlier detection to the raw waveform data before bandpass filtering[28]. We extracted a 2 min envelope from the original velocity trace whenever its value surpassed three standard deviations above the mean within a 10 min window. This action effectively eliminated signal interference caused by the majority of brief events. However, interference from prolonged teleseismic waves resulting from significant subduction earthquakes persisted, necessitating the application of a two-window (20 min) moving minimum to filter out this remaining contamination from the data stream[28].

## Data normalization
We used z-score normalization applied in log space to eliminate disparate magnitude and range effects arising between the different data streams[28]. This produces normalized distributions of data stream values, which prevents one data type from dominating feature selection. It also improves comparisons between volcanoes where stations are located at different distances from the vent. In log space, distributions exhibit central tendencies and do not display excessively large asymmetries (Fig. S6). Using the entire signal record for normalization can lead to information leakage because it incorporates future data into the scaling process, potentially biasing the model's performance estimates. However, we checked the effect of removing the small amount of pre-eruption data from normalization and the

effects are imperceptible. An alternative normalization approach could be to use reduced displacement[52,53].

## Feature calculation
Data time series are divided into 48-hour windows, with each window consisting of 288 samples, each sample 10 min in length. During model training, we allow adjacent windows to overlap by 36 h (75% of their length) which increases the likelihood that a precursor pattern is fully captured within at least one window rather than split by window boundary. During model forecasting, adjacent windows overlap almost entirely, except for a one sample (10-minute) shift forward in time. For each data stream in each window, we calculate over 700 time series features using the Python package "tsfresh[54]". These features include distribution measures (mean, standard deviation, number of peaks), correlation measures (lags, Fourier coefficients), linearity (slope and intercept of linear regressors), and other information measures (entropy, energy, nonlinear scores). All feature values are stored in a matrix, where individual columns represent different features, and rows represent the end-time index of each window. Our choice to use a 48-hour window focuses on detecting short-term fluctuations in volcano state. This ultimately derives from the intent of this analysis, which is to provide information on short-term changes in hazards that would be useful for responsive volcanic monitoring. We affirm that shorter or longer time windows might also be useful and could be explored in other studies. However, previous studies[28,50] explored a range of windows between 12 h and 5 days at Whakaari and found that the choice did not greatly impact forecast accuracy at that volcano. We further note that some volcanoes show meaningful unrest on longer timescales[55].

## Labeling
We define a classification problem that prioritizes algorithm attention on signals occurring in the 48 h before eruptions, called the "look forward" window. All feature windows whose end time occurs 48 h or less before the date of an eruption onset are labeled "1", while the remaining windows are labeled as "0". For training windows that overlap by 75%, this results in about 164 label-1 windows (pre-eruptive) and more than 50,000 label-0 windows (non-eruptive). The large imbalance is handled by randomly discarding non-eruptive "0" windows until the dataset is balanced. The process of randomly discarding non-eruptive "0" windows until the dataset is balanced is repeated multiple times to minimize the influence of random selection when removing non-eruptive windows during downsampling.

## Feature ranking
For a given down-sampled window subset, feature values are subdivided into groups corresponding to label-0 and label-1. The importance of each feature is then determined by whether there is a statistically significant difference between the two distributions of feature values for each label class. Significance is assessed using the Mann-Whitney U test, which evaluates for a possible difference in median between two distributions. Features are retained as significant if the output $p$-value from the test falls below an adjusted Benjamini-Yekutieli threshold[50,56], which controls for a fixed false discovery rate of 5% when conducting many statistical tests. Features are then ranked (by lowest $p$-value) and passed on for model training.

## Training
First, out-of-sample volcano data are set aside for testing and not used during feature selection or model training. We use a random forest algorithm to solve the classification problem. Each model consists of up to 200 individual decision trees, with each tree having been trained on a different subset of down-sampled windows. Thus, all non-eruptive data appears, at a very low frequency, across

the training set[50]. This means that within the training set used for the random forest algorithm, non-eruptive data instances are represented sparsely. Models are trained using the scikit-learn Python library[57]. The random forest classification model we implement has been preferred as it was previously found to outperform other classifiers on this problem[50].

## Pseudo-prospective forecasting

Forecasts are only constructed for data withheld from feature selection and model training steps, simulating real-time conditions by ensuring the test data remains unseen. This approach approximates but does not fully replicate prospective forecasting conditions, offering a more realistic assessment of performance than a hindcast, which includes test data in the training set. For the test data, feature time series are calculated using overlapping 48-hour windows that each advance the previous by 10 min and hence overlap at 287 out of 288 points. This is intended to imitate real-time volcano monitoring where data is interpreted as it is received, i.e., at 10 min intervals. Features from each subsequent window are passed to the trained random forest model. The binary outputs of each decision tree are averaged to produce a value ranging from 0 to 1, referred to as the consensus of the forecast model.

## Volcano training pools

Volcanoes were divided into distinct groups or "pools" based on their reported eruption types. We used three specific pools in this study (Fig. 1b). The magmatic pool comprised volcanoes in Alaska (Redoubt, Augustine, Veniaminof, Pavlof, Great Sitkin, Semisopochnoi, Okmok), Kamchatka, Russia (Bezymianny), and Washington, USA (Mt. St. Helens). These volcanoes were included due to their shared tectonic origin, and tendency to produce magmatic eruptions. The phreatic pool included New Zealand volcanoes (Whakaari, Ruapehu, Tongariro), as well as Mt. Ontake in Japan (Yamaoka et al., 2016), Copahue in Chile/Argentina[51], and Kawah Ijen in Indonesia[51]. These volcanoes all have well-developed hydrothermal systems. Finally, a world pool comprising all the volcanoes was included.

## Forecast evaluation

The purpose of forecast testing is to determine overall accuracy, including the false positive rate (incorrectly forecasted that an eruption will occur), false negative rate (failing to forecast an eruption), and the correlation between forecasted probabilities and actual outcomes over long time periods. Accuracy can be tested under pseudo-prospective and prospective conditions with the latter needing real-time or future data to assess a frozen model —a finalized model that remains unchanged during evaluation[28]. Pseudo-prospective forecast performance is not guaranteed to replicate real-time prospective performance.

In machine learning, cross-validation is used to evaluate model performance as well as its ability to generalize to new data or contexts. Leave-One-Out Cross-Validation (LOOCV) trains a model using all samples (volcanoes or eruptions) in a dataset except for one, and then using the excluded sample as a validation set to measure model performance. This is repeated for each sample in the dataset and the performance of all models is combined for an overall estimate of model performance. Here, we applied LOOCV at the volcano level when constructing generalized ML forecasts. We applied LOOCV at the individual eruption level when making comparisons to tailored ML forecasts (Fig. 1c). See Fig. S7 for an alternative scheme.

## Receiver operating characteristic (ROC) curves

ROC curves measure the ability of machine learning classification models to correctly classify whether an eruption will or will not occur. All ROC curves shown here are calculated for out-of-sample forecasts.

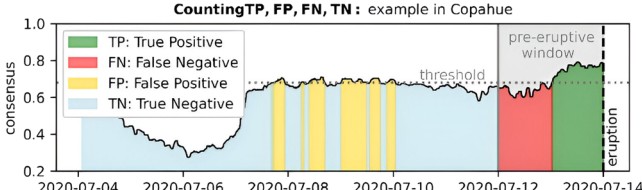

**Fig. 6 | The figure shows how the consensus time series, which is the output of the eruption forecasting classifier model, is used to count true positives (TP), false positives (FP), true negatives (TN), and false negatives (FN) using a threshold and a pre-defined eruption window.** Threshold: A horizontal line between 0 and 1 on the model output axis. In this study, we tested for 100 thresholds equally distributed between 0 and 1. Pre-eruptive window: A shaded area on the time axis representing the time period before the eruption consider as pre-eruptive (corresponding to the training window). True positive (TP): Any model output that surpasses the threshold after the start of the pre-eruptive window and before the eruption is considered a TP. These are correctly predicted eruptions. False negative (FN): Any model output that does not surpass the threshold between the beginning of the pre-eruptive window and the first time it surpasses the threshold (if it ever does) is considered an FN. These are missed eruptions. False positive (FP): Any model output that surpasses the threshold outside the pre-eruptive window is considered an FP. These are incorrectly forecasted eruptions. True negative (TN): Any model output that does not surpass the threshold outside the pre-eruptive window. These are correctly classified non-eruptive data. Note: The selection of the threshold value can significantly impact the number of TP, FP, FN, and TN.

The ROC curve tracks how true positive (sensitivity) and false positive rate (1-specificity) co-evolve for a trigger threshold that is swept from 0 to 1. Hence, they provide a single evaluation of forecast independent of threshold choice. The ROC curve is quantified by its area under the curve (AUC), with values approaching 1 indicating perfect performance.

Here, we constructed ROC curves according to the following steps (See Fig. 6): (1) a threshold value is selected, (2) the threshold is applied to the forecast to create a binary output (1=consensus exceeds threshold, 0=consensus is equal to or below threshold); (3) a true positive (TP) or true negative (TN) is scored in each window that the forecast model agrees with the out-of-sample data label: both "1" for TP or both "0" for TN (Fig. 6); (4) a false positive (FP) is scored when the forecast model outputs a "1" but this does not match a corresponding "0" of the label vector; a false negative (FN) is scored for all other mismatches (Fig. 6); (5) the true positive rate (TPR) and false positive rate (FPR) are calculated as TPR = TP / (TP + FN) and FPR = FP / (FP + TN); (6) steps (1-5) are repeated using 100 evenly spaced thresholds between 0 and 1 with the set of matching pairs forming the ROC curve; finally (7) AUC is calculated as the integral of the ROC curve.

## Seismic amplitude forecast

To benchmark the generalized ML forecasts, we construct a simple forecast based on seismic amplitude. An RSAM forecast consensus is calculated by transforming the raw RSAM record calculated in this study for each volcano to an equivalent percentile value, e.g., an RSAM value at the median would correspond to an RSAM forecast consensus of 0.5. The percentile evaluation is done on a volcano-by-volcano basis and, as it assumes knowledge of future extreme maximum values, it is not pseudo-prospective. Although this means the RSAM model has an information advantage over generalized ML forecasts, we deemed this acceptable as it is here used only as a reference case. The RSAM model was further advantaged by testing three variations: direct use of its output on a 10 min basis, or a rolling median using 6 h or 2 days. We selected the version with the highest accuracy, which turned out to be the 6-hour rolling median, for comparison to the ML forecasts.

## Potential avenues for enhancing methods

To enhance accuracy, avenues for improvement include expanding precursory analysis by considering longer time periods (multi-time scale approaches; Ardid et al., 2023; Ardid et al., 2024) and incorporating additional data types such as gas emission rate[58,59], thermal anomalies[60,61], and magnetotelluric data[62–64], and from extended global datasets[65] Also, application to geysers monitored with seismic data[66,67] given their frequent eruptive nature and lower time between eruption offers an opportunity to improve these methods with more balanced datasets between eruptive and non-eruptive data for the ML classification problem.

Another avenue for improvement is on the outlier detection algorithm, which is intended to filter out regional earthquakes, and its calibration was based on data specifically gathered from regional earthquakes[28]. However, it filters out all earthquakes without discriminating, including VTs—this is because our research interest is in online real-time algorithms that can operate without human assistance (e.g., classifying VT earthquakes)—future work could include adding automated discrimination of VTs from regional earthquake[68–70].

## Data availability

The dataset for most volcanoes in the catalog, pre-processed as amplitude measurements sampled every 10 min, is included in the Supplementary Data 1. Raw waveform data for New Zealand volcanoes can be downloaded from GEONET, and for Alaskan volcanoes from IRIS. Both datasets are operable as clients through the FDSN webservice (https://www.fdsn.org/networks/). If you utilize the provided seismic data, please cite this article accordingly. Users are also expected to adhere to the non-commercial nature of the provided datasets and materials.

## Code availability

The codes used in this study are provided in the Supplementary Code 1. The repository includes the main library for forecasting, named Puia (meaning "volcanoes" in Māori), along with scripts to implement a generalized forecaster, test a simple RSAM forecaster, and compute ROC curves. The code is released under the Creative Commons Attribution-NonCommercial (CC BY-NC) License, allowing free use, modification, and distribution of the software for non-commercial purposes, provided the original license and copyright notice are included. Any third-party use of this software for commercial purposes is strictly prohibited without explicit permission from the corresponding author. Users are also expected to adhere to the non-commercial nature of the provided datasets and materials. This software is not guaranteed to be entirely free of bugs or errors. Minor issues may exist, but they are expected to have only a marginal impact on accuracy and performance. If you discover any bugs or errors, we encourage you to report them by contacting the corresponding author. Please note that this software is not designed to serve as an Application Programming Interface (API) for building custom volcano eruption forecast models, nor is it particularly user-friendly for individuals new to Python or machine learning. However, if you wish to adapt this model for another volcano or a different use case, we encourage you to do so. We welcome inquiries regarding the best way to proceed and are happy to provide guidance when possible. The authors acknowledge the preprint of this manuscript, which is available online at https://doi.org/10.21203/rs.3.rs-3483573/v1[71].

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

## Acknowledgements
We acknowledge GEONET (New Zealand) and AVO (Alaska Volcano Observatory, USA) seismic monitoring systems for supplying free access to their volcanic seismic data. We wish to thank NZ MBIE UOAX1913 (Transitioning Taranaki to a Volcanic Future) for support of AA, DD, and SJC during this study. We wish to thank NZ MBIE E7774 (Adapting to climate change through stronger geothermal enterprises') for support of AA, DD, and SJC during this study. CC acknowledges the funding from the Fonds De La Recherche Scientifique - FNRS (CalderaSounds project), the Fondation Wiener Anspach, and the Wel Research Institute (Geo4D project). LC thanks the support of the 'Programa de Riesgo Sísmico' (PRS, Actividades de Interés Nacional, Universidad de Chile). SP and OL were supported by the New Zealand Ministry of Business, Innovation & Employment (MBIE) through the GNS Science Hazard & Risk Management program (Strategic Science Investment Fund, contract C05X1702). Support from the U.S. Geological Survey under Cooperative Agreement No. G21AC10384 is acknowledged. Silvio De Angelis was supported by "Progetto INGV Pianeta Dinamico" -Sub-project VT_DYNAMO 2023- code CUP D53J19000170001 - funded by MIUR ("Fondo Finalizzato al rilancio degli investimenti delle amministrazioni centrali dello Stato e allo sviluppo del Paese", legge 145/2018). We thank the Icelandic Meteorological Office (IMO) and Dr Kristín Jónsdóttir for providing data for the Eyja. volcano. We thank the Observatorio Volcanológico de los Andes del Sur (OVDAS-SERNAGEOMIN) and Dr Francisco Delgado for providing the seismological data of the Caulle volcano through the Portal de Transparencia system, which is freely accessible through their website. We thank the Dr Ivan Melchor and 'Instituto de Investigación en Paleobiología y Geología' (UNRN-CONICET), Argentina, for providing the seismological data of the Copahue volcano. We thank Dr Yuta Maeda and the Nagoya University, Japan, for providing data of the Mt Ontake. We also thank Dr Yasua Ogawa for its support for collecting this data. The authors thank the British Geological Survey for providing access to the seismic data for Soufrière Hills Volcano. We present and analyze the pre-eruptive performance of models exclusively for volcanoes where data were collected through open access, which are 12 volcanoes in the US and New Zealand. This is due to observatories sensitivities and shared data agreements. However, it's important to note that the models are trained using all available data for each respective pool.

## Author contributions
A.A. and D.D. made equal contributions to this manuscript by developing the machine learning pipeline, which includes the design and programming, and writing of the manuscript. C.C., T.G., B.K., D.R., and S.C. contributed to the analysis, interpretation and contributed to manuscript development. C.M., L.C., S.R., S.P., J.P., M.M., A.M., Y.C., S.A., O.L., R.C., contributed by sharing data and to the manuscript.

## Competing interests
The authors declare no competing interests.
