## [Transparent Peer Review file · Nature Communications]

Ergodic Seismic Precursors and Transfer Learning for Short Term Eruption Forecasting at Data Scarce Volcanoes

Corresponding Author: Dr Alberto Ardid

Version 1:

Reviewer comments:

Reviewer #1

(Remarks to the Author)

In their submission, Ardid et al. Show that a random forest model can be trained to forecast volcano eruptions from features of single seismic stations.

I initially reviewed this paper for Nature (reviewer 1). My main comments were:

- 1) That this work builds a lot on a previous paper by the authors (<https://www.nature.com/articles/s41467-022-29681-y>), in which the most important geophysical insights were already laid out in my opinion, with this current paper being only about the addition of ML on top of their previous work.
- 2) That there is a lack of comparison with existing eruption forecasting systems, and without a baseline to compare against, one cannot tell if the authors' method provides any advantage over existing methods.

I cannot recommend publication because my main concerns still hold: the addition of ML does not provide on its own the kind of new geophysical insights that one would expect from a Nature Communications paper, and the author have not addressed the lack of comparison with other forecasting systems.

(Remarks on code availability)

Reviewer #2

(Remarks to the Author)

Comments to the manuscript entitled "Generalized eruption forecasting models using Machine Learning trained on seismic data from 24 volcanoes" from Ardid et al.

General comments.

The paper by Ardid et al. is a very interesting and appropriate approach to obtaining pre- and syn-eruptive models of active volcanoes using seismic data. Their results may in the near future be used as input models in future early warning protocols for volcanic eruptions. They analyse a very important database, due to the amount of data, the time period of the data and the variety of eruptive processes, in order to represent a sample very close to the generalisation of the process. They use a modern methodology that is rarely used in the field of volcanic seismology, which also adds a highly relevant innovative aspect to the impressive database that perfectly fits the profile of the journal. But there are some previous works recently published, which I am sure the authors of this manuscript are aware of and which have not been mentioned, I assume by mistake. I am referring mainly to the following three papers that should be cited to provide a true state of the art on the subject:

Rey-Devesa, P., Prudencio, J., Benítez, C., Bretón, M., Plasencia, I., León, Z., ... & Ibáñez, J. M. (2023). Tracking volcanic explosions using Shannon entropy at Volcán de Colima. *Scientific Reports*, 13(1), 9807.

Rey-Devesa, P., Benítez, C., Prudencio, J., Gutiérrez, L., Cortés-Moreno, G., Titos, M., ... & Ibáñez, J. M. (2023). Volcanic early warning using Shannon entropy: Multiple cases of study. *Journal of Geophysical Research: Solid Earth*, 128(6),

e2023JB026684.

Rey-Devesa, P., Carthy, J., Titos, M., Prudencio, J., Ibáñez, J. M., & Benítez, C. (2024). Universal machine learning approach to volcanic eruption forecasting using seismic features. *Frontiers in Earth Science*, 12, 1342468.

As a senior researcher in this field, it is a great joy for me that finally this editorial group, the editors and the previous reviewers have understood that this type of work should be published in these high-impact journals given their scientific and social importance. Volcanic eruptions represent hazards to humanity in several ways; they are one of the fundamental sources of Climate Change gases and also directly threaten people and cities. The development of a generalized/universal and useful tool for early warning of volcanic eruptions that will help to the humanity to more effectively manage volcanic crises is clearly needed, and this manuscript is a great advance in this field.

The manuscript is very well written, its English is perfectly understandable, as is to be expected from the nationality of most of the authors, and it clearly and efficiently covers all the aspects necessary for publication in its current state.

The version I am reviewing corresponds to at least a second revision that the authors have made in response to comments from at least 5 different reviewers. Each of these reviewers has made their own comments, some of which are inconsistent with the others. Others are really comments based on subjective desires or, let's say, philosophical considerations of the reviewers rather than purely scientific comments related to the subject of the research. Other comments have been really pertinent and focused on trying to clarify the manuscript, not on improving the research, because the underlying research is excellent.

However, the authors have been able to respond clearly and efficiently to each of the objections and comments made by the reviewers. In this sense, this version covers everything previously required in an efficient, clear and realistic manner. In this sense, the text should, in my opinion, be published as is because the authors' effort has been impressive in satisfying all the requirements.

As a reviewer, author, expert, project manager and other research activities, I believe that one should not get into an endless cycle in which new reviewers give their opinion on a job well done. Of course, we could all give our opinion!!! (in fact, I have asked to include three references that I think are important!!!), but for the sake of fair, efficient, clear and effective science, this manuscript should not be revised further and deserves to be published as soon as possible.

Yours sincerely
Jesús M. Ibáñez

(Remarks on code availability)
All results are reproducible

Reviewer #3

(Remarks to the Author)

This manuscript presents a method to improve volcanic eruption forecasting using machine learning techniques exploiting available monitoring seismic signals, also with the aim to export this approach to volcanoes lacking instrumental data.

While I am not an expert in machine learning, I find that the manuscript has a rigorous, solid and overall convincing approach and its general outcome is original and sound.

It may be an important and interesting contribution to volcanology and eruption forecasting.

I only have a few minor points, listed below, that I suggest should be improved.
More detailed annotations are reported in the attached file.

1) At several places throughout the first part of the manuscript you mention "precursors". Are these really precursors? Which is your definition of a precursor? What makes these indicators precursors? This is not trivial, as I suspect that most of these "precursors" are just given from granted. Many claimed precursors in the literature are actually poorly defined, or defined only on retrospective, which makes them potentially weak. I would like to see some convincing explanation for their mentioning here.

2) A few references are missing (see attached file).

3) The role of the opening or closing of the conduit just before the eruption seems at times to be not adequately considered in considering the observed signals.

4) I find the discussion too much synthetic and poorly informative, so that one does not really understand the conveyed message, and how justified this may be.

Which is the main message? ergodicity?

So, which is the general sequence of seismic events we should expect according to ergodicity?

I suggest to describe your message in more detail referring to the analyzed pools.

(Remarks on code availability)

Review form *Nature Communications*

Editorial comment is normal font.

Our direct response is italicised and in blue.

REVIEWER COMMENTS

Reviewer #1 (Remarks to the Author):

In their submission, Ardid et al. Show that a random forest model can be trained to forecast volcano eruptions from features of single seismic stations.

I initially reviewed this paper for Nature (reviewer 1). My main comments were:

- 1) That this work builds a lot on a previous paper by the authors (<https://www.nature.com/articles/s41467-022-29681-y>), in which the most important geophysical insights were already laid out in my opinion, with this current paper being only about the addition of ML on top of their previous work.
- 2) That there is a lack of comparison with existing eruption forecasting systems, and without a baseline to compare against, one cannot tell if the authors' method provides any advantage over existing methods.

I cannot recommend publication because my main concerns still hold: the addition of ML does not provide on its own the kind of new geophysical insights that one would expect from a Nature Communications paper, and the author have not addressed the lack of comparison with other forecasting systems.

We appreciate your comments. Below are our responses to address the concerns.

Overlap with Our Previous Work. *Our prior paper identified seismic features correlated with pre-eruptive activity in six volcanoes. This study, however, progresses from correlation to prediction by rigorously testing a machine learning model designed for eruption forecasting. Unlike the previous study, which only suggested correlations, this work focuses on generalizability by validating the model on 24 volcanoes outside the training set, using out-of-sample forecasts evaluated through ROC curves and AUC scores. Additionally, we benchmarked the model against RSAM-based monitoring, demonstrating its capability to detect latent eruption precursors and extending its application within volcanic monitoring.*

Significance of This Study and Comparison. *Our approach builds on analogue forecasting, which typically uses data from well-documented volcanoes to inform less-studied systems. This study adopts a data-driven approach that assesses seismic patterns across multiple volcanoes, addressing data scarcity through machine learning (ML) rather than expert judgment. By using ML to identify patterns within seismic data, the model reduces reliance on subjective criteria, supporting generalization of precursor patterns across diverse systems, including volcanoes with limited eruption histories. This adjustment allows for objective, cross-system pattern recognition, extending analogue forecasting to data-scarce and intermittently erupting volcanoes.*

Comparison with Existing Forecasting Systems. *We recognize the importance of placing our model in the context of existing forecasting systems. However, direct comparisons are challenging, as many established systems employ proprietary methodologies. Instead, we benchmarked our ML model against RSAM, a widely used tool in seismic amplitude monitoring with a high sampling rate*

suitable for reflecting volcanic activity. Although calibrated for RSAM, our ML model consistently performed well in out-of-sample forecasts, effectively identifying eruption-precursor signals. As noted in the text: "This comparison is not intended to diminish RSAM's value in volcano monitoring, consistently demonstrated at observatories worldwide. Instead, we highlight that latent patterns within RSAM and similar seismic measurements are effectively extracted here to deliver monitoring and forecast insights." Thus, while RSAM remains a valuable data source, our ML approach captures subtle precursor patterns for improved eruption forecasting and serves as a complementary tool to traditional RSAM-based monitoring.

Generalized Model Performance and Ergodic Characteristics: *Our study introduces ergodicity in volcanic eruption forecasting, where similarities across multiple volcanoes allow for generalized model predictions at target volcanoes beyond the training set, thus reducing dependence on site-specific historical data. As described in the text: "The generalized models explored here exploit ergodicity in volcanic systems at the time series feature level... This is a property of physical systems whereby observations drawn from a large enough (volcano) ensemble contain sufficient information to approximate future behavior at a target (volcano) outside the subset."*

Reviewer #2 (Remarks to the Author):

Comments to the manuscript entitled "Generalized eruption forecasting models using Machine Learning trained on seismic data from 24 volcanoes" from Ardid et al.

General comments.

The paper by Ardid et al. is a very interesting and appropriate approach to obtaining pre- and syn-eruptive models of active volcanoes using seismic data. Their results may in the near future be used as input models in future early warning protocols for volcanic eruptions. They analyse a very important database, due to the amount of data, the time period of the data and the variety of eruptive processes, in order to represent a sample very close to the generalisation of the process. They use a modern methodology that is rarely used in the field of volcanic seismology, which also adds a highly relevant innovative aspect to the impressive database that perfectly fits the profile of the journal. But there are some previous works recently published, which I am sure the authors of this manuscript are aware of and which have not been mentioned, I assume by mistake. I am referring mainly to the following three papers that should be cited to provide a true state of the art on the subject:

Rey-Devesa, P., Prudencio, J., Benítez, C., Bretón, M., Plasencia, I., León, Z., ... & Ibáñez, J. M. (2023). Tracking volcanic explosions using Shannon entropy at Volcán de Colima. *Scientific Reports*, 13(1), 9807.

Rey-Devesa, P., Benítez, C., Prudencio, J., Gutiérrez, L., Cortés-Moreno, G., Titos, M., ... & Ibáñez, J. M. (2023). Volcanic early warning using Shannon entropy: Multiple cases of study. *Journal of Geophysical Research: Solid Earth*, 128(6), e2023JB026684.

Rey-Devesa, P., Carthy, J., Titos, M., Prudencio, J., Ibáñez, J. M., & Benítez, C. (2024). Universal machine learning approach to volcanic eruption forecasting using seismic features. *Frontiers in Earth Science*, 12, 1342468.

As a senior researcher in this field, it is a great joy for me that finally this editorial group, the editors and the previous reviewers have understood that this type of work should be published in these high-impact journals given their scientific and social importance. Volcanic eruptions represent hazards to humanity in several ways; they are one of the fundamental sources of Climate Change gases and also directly threaten people and cities. The development of a generalized/universal and useful tool for early warning of volcanic eruptions that will help to the humanity to more effectively manage volcanic crises is clearly needed, and this manuscript is a great advance in this field.

The manuscript is very well written, its English is perfectly understandable, as is to be expected from the nationality of most of the authors, and it clearly and efficiently covers all the aspects necessary for publication in its current state.

The version I am reviewing corresponds to at least a second revision that the authors have made in response to comments from at least 5 different reviewers. Each of these reviewers has made their own comments, some of which are inconsistent with the others. Others are really comments based on subjective desires or, let's say, philosophical considerations of the reviewers rather than purely scientific comments related to the subject of the research. Other comments have been really pertinent and focused on trying to clarify the manuscript, not on improving the research, because the underlying research is excellent.

However, the authors have been able to respond clearly and efficiently to each of the objections and comments made by the reviewers. In this sense, this version covers everything previously required in an efficient, clear and realistic manner. In this sense, the text should, in my opinion, be published as is because the authors' effort has been impressive in satisfying all the requirements.

As a reviewer, author, expert, project manager and other research activities, I believe that one should not get into an endless cycle in which new reviewers give their opinion on a job well done. Of course, we could all give our opinion!!! (in fact, I have asked to include three references that I think are important!!!), but for the sake of fair, efficient, clear and effective science, this manuscript should not be revised further and deserves to be published as soon as possible.

Yours sincerely
Jesús M. Ibáñez

We appreciate the reviewer's supportive assessment of our manuscript. It is encouraging to receive positive feedback on the relevance of this approach and its possible impacts on volcanic eruption forecasting.

In response to your suggestion, we have reviewed the three references mentioned. We agree that these bring important context in describing the state of the art and hence have included them in the revised manuscript.

Reviewer #3 (Remarks to the Author):

This manuscript presents a method to improve volcanic eruption forecasting using machine learning techniques exploiting available monitoring seismic signals, also with the aim to export this approach to volcanoes lacking instrumental data.

While I am not an expert in machine learning, I find that the manuscript has a rigorous, solid and overall convincing approach and its general outcome is original and sound.

It may be an important and interesting contribution to volcanology and eruption forecasting.

I only have a few minor points, listed below, that I suggest should be improved.
More detailed annotations are reported in the attached file.

Thank you for your constructive feedback and assessment of our manuscript.

1) At several places throughout the first part of the manuscript you mention "precursors". Are these really precursors? Which is your definition of a precursor? What makes these indicators precursors? This is not trivial, as I suspect that most of these "precursors" are just given from granted. Many claimed precursors in the literature are actually poorly defined, or defined only on retrospective, which makes them potentially weak. I would like to see some convincing explanation for their mentioning here.

Definition of Precursors: We agree that the term "precursor" deserves further clarification. In our study, we use the term to refer to features in seismic data that are statistically more frequent in the days prior to eruptions across multiple volcanoes compared to their occurrence in background data. We have included a clearer definition in the manuscript to avoid ambiguity and ensure consistency in how we describe these signals.

2) A few references are missing (see attached file).

We have incorporated the references mentioned.

3) The role of the opening or closing of the conduit just before the eruption seems at times to be not adequately considered in considering the observed signals.

We agree that these processes can influence signal patterns. Open and closed conduit eruptions might be another good way to subdivide different eruption pools and hence find greater model effectiveness. Although we have not done so here, we now specifically highlight this possibility through the manuscript.

4) I find the discussion too much synthetic and poorly informative, so that one does not really understand the conveyed message, and how justified this may be.

Which is the main message? ergodicity? So, which is the general sequence of seismic events we should expect according to ergodicity? I suggest to describe your message in more detail referring to the analyzed pools.

Our discussion attempts to position the main scientific finding (ergodicity in eruption precursors) in response to a key societal challenge (not enough data for all volcanoes) while paying due credit to its limitations. We have added a clarifying statement at the beginning of this section that we hope brings useful context to the individual points discussed in subsequent paragraphs.

We have elaborated on how our results support the concept of applying insights from one volcano to others with similar eruption patterns. Additionally, we provided more detailed on the model performance over individual volcanoes. This will help communicate how our findings contribute to understanding and forecasting eruption processes through ergodic models.